# Pyrimidine-Based Push–Pull Systems with a New Anchoring Amide Group for Dye-Sensitized Solar Cells

Egor V. Verbitskiy [1,2,*], Alexander S. Steparuk [1], Ekaterina F. Zhilina [1], Viktor V. Emets [3], Vitaly A. Grinberg [3], Ekaterina V. Krivogina [4], Sergey A. Kozyukhin [4,5], Ekaterina V. Belova [6], Petr I. Lazarenko [7], Gennady L. Rusinov [1,2], Alexey R. Tameev [3], Jean Michel Nunzi [8,9] and Valery N. Charushin [1,2]

1   I. Postovsky Institute of Organic Synthesis, Ural Branch of the Russian Academy of Sciences, S. Kovalevskaya Str., 22, 620108 Ekaterinburg, Russia; steparuk96@mail.ru (A.S.S.); efzhilina@ios.uran.ru (E.F.Z.); g.rusinov@mail.ru (G.L.R.); valery-charushin-562@yandex.ru (V.N.C.)
2   Department of Organic and Biomolecular Chemistry, Chemical Engineering Institute, Ural Federal University, Mira St. 19, 620002 Ekaterinburg, Russia
3   A. N. Frumkin Institute of Physical Chemistry and Electrochemistry, Russian Academy of Sciences, 119071 Moscow, Russia; victoremets@mail.ru (V.V.E.); vitgreen@mail.ru (V.A.G.); altam2001@mail.ru (A.R.T.)
4   N. S. Kurnakov Institute of General and Inorganic Chemistry, Russian Academy of Sciences, 31, Leninsky Pr., 119991 Moscow, Russia; ekaterina3141@mail.ru (E.V.K.); sergkoz@igic.ras.ru (S.A.K.)
5   Moscow Institute of Physics and Technology, National Research University, 9 Institutskiy per., 141701 Dolgoprudny, Russia
6   Department of Chemistry, Moscow State University, 119991 Moscow, Russia; catrine2@mail.ru
7   Institute of Advanced Materials and Technologies, National Research University of Electronic Technology, 124498 Moscow, Russia; lpi@org.miet.ru
8   Nanomaterials Research Institute, Kanazawa University, Kanazawa 920-1192, Japan; nunzijm@queensu.ca
9   Department of Physics, Engineering Physics and Astronomy, Queen's University, Kingston, ON K7L-3N6, Canada
*   Correspondence: verbitskye@yandex.ru

**Abstract:** New donor–π–acceptor pyrimidine-based dyes comprising an amide moiety as an anchoring group have been designed. The dyes were synthesized by sequential procedures based on the microwave-assisted Suzuki cross-coupling and bromination reactions. The influence of the dye structure and length of π-linker on the photophysical and electrochemical properties and on the photovoltaic effectiveness of dye-sensitized solar cells was investigated. An increase in efficiency with a decrease in the length of π-linker was revealed. The **D1** dye with only one 2,5-thienylene-linker provided the highest power conversion efficiency among the fabricated dye sensitized solar cells.

**Keywords:** pyrimidine; triphenylamine; amide group; dye-sensitized solar cells





## 1. Introduction

Dye-sensitized solar cells (DSSCs) are one of the budding kinds of devices for the future large-scale production of electricity from renewable energy sources [1–3]. In these cells, the sensitizer appears as one of the main structural elements that harvest solar energy and transforms it into electric power. In general, organic sensitizers consist of a donor–(π-linker)–acceptor (D–π–A) system [2]. Cyanoacetic acid is the most famous acceptor, which is commonly used as an anchoring group for DSSCs [1]. However, the cyanoacetic group controls the redox properties, limiting the ability to tune the HOMO–LUMO level. Therefore, an investigation of alternative anchoring groups is unequivocally required to develop the next generation of stable organic sensitizers. Due to the recent huge interest in DSSCs technology, numerous anchoring groups have appeared and been evaluated [4].

The azaheterocyclic fragment is a well-known electron-accepting moiety, which is synthetically straightforward to introduce, making it an interesting candidate in D–π–A sensitizers for DSSCs [5–9]. Nevertheless, to date, there have been only few examples of azine-functionalized push–pull systems for DSSC applications, and the reported efficiencies

with such dyes have been low. Particularly, for the first time, we recently synthesized a series of novel push-pull dyes comprising pyrimidine as the anchor. Unfortunately, it was found that DSSCs based on these dyes had unimpressive power conversion efficiencies (PCE) up to 1% [8,9].

This paper is a further enlargement of our research, which is focused on synthetic procedures for the modification of pyrimidine-based push–pull systems for photovoltaic applications. Here we compared three pyrimidine dye molecules **D1**–**D3** (Figure 1) in which one or two triphenylamine groups are used as donor parts, conjugated thiophene linkers with different structures as π-system and amide group as an anchoring unit. By comparing the photophysical and electrochemical properties and device performance, the relationship between the structures and properties of these three dyes has been obtained.

**Figure 1.** Chemical structures of **D1**–**D3**.

## 2. Experimental

Since we have already repeatedly described general data on the used analytical equipment and methods [10–12], they are presented in the "**General Information**" section (see Supplementary Materials).

The general procedure for the Suzuki cross-coupling reactions for the synthesis of compounds 6, 8, 10, 12 and 14:

A solution of $K_2CO_3$ (346 mg, 2.5 mmol (generally) or 691 mg, 5.0 mol in the reaction with **13**) in $H_2O$ (4 mL) was added to a mixture of the corresponding bromo-substituted pyrimidine (1.0 mmol), arylboronic derivative (**5**, **7** or **11**) (1.2 mmol (generally) or 2.4 mol in the reactions with **13**) and $Pd(PPh_3)_4$ (29 mg, 5 mol% (generally) or 58 mg, 10 mol% in the reactions with **13**) in 1,4-dioxane (8 mL). The reaction mixture was degassed and irradiated in a microwave apparatus at 165 °C (250 W) for 30 min. The reaction mixture was cooled, filtered and dissolved in a mixture of EtOAc and water (1:1, 50 mL), and the organic layer was separated. The aqueous layer was extracted with EtOAc (2 × 25 mL). The combined organic extracts were dried with $MgSO_4$ and the solvents evaporated. Purification by silica gel column chromatography with EtOAc/hexane (gradient from 1:4 to 1:1, *v/v*) as an eluent to afford the desired cross-coupling products.

**4-(4-(5-(4-(Diphenylamino)phenyl)thiophen-2-yl)pyrimidin-5-yl)benzonitrile (6)**. The Suzuki cross-coupling reaction of 4-(4-(5-bromothiophen-2-yl)pyrimidin-5-yl)benzonitrile (**6**) with 4-(diphenylamino)phenylboronic acid pinacol ester (**5**) (which has been performed according to the general procedure) gave after purification by column chromatography 380 mg (75%) of **6** as a yellow solid. Melting point: 164–166 °C. [1]H NMR (500 MHz, CDCl₃) δ 9.14 (s, 1H), 8.49 (s, 1H), 7.81 (d, *J* = 7.7 Hz, 2H), 7.56 (d, *J* = 7.8 Hz, 2H), 7.44 (d, *J* = 8.3 Hz, 2H), 7.32–7.22 (m, 5H), 7.12 (d, *J* = 7.9 Hz, 3H), 7.04 (td, *J* = 18.2, 17.3, 5.7 Hz, 5H), 6.69 (d, *J* = 4.1 Hz, 1H). [13]C NMR (126 MHz, CDCl₃) δ 158.0, 157.5, 1560,

149.9, 148.4, 147.1, 141.4, 138.6, 133.0, 132.2, 130.1, 129.4, 128.8, 126.8, 126.7, 124.9, 123.6, 123.1, 122.7, 118.3, 112.9. Calcd. for $C_{33}H_{22}N_4S$ (506.62): C, 78.23; H, 4.38; N, 11.06. Found: C, 78.11; H, 4.56; N, 11.01. $\nu$ (DRA, cm$^{-1}$) 2228 (s, C≡N)

**4-(4-([2,2′-Bithiophen]-5-yl)pyrimidin-5-yl)benzonitrile (8)**. The Suzuki cross-coupling reaction of 4-(4-(5-bromothiophen-2-yl)pyrimidin-5-yl)benzonitrile (**6**) with 2-thienylboronic acid (**7**) (which has been performed according to the general procedure) gave after purification by column chromatography 287 mg (83%) of **8** as a yellow solid. Melting point: 168–170 °C. $^1$H NMR (500 MHz, CDCl$_3$) δ 9.14 (s, 1H), 8.50 (s, 1H), 7.84–7.80 (m, 2H), 7.57–7.54 (m, 2H), 7.28 (dd, *J* = 5.1, 1.2 Hz, 1H), 7.25 (dd, *J* = 3.6, 1.2 Hz, 1H), 7.04 (dd, *J* = 5.1, 3.6 Hz, 1H), 6.95 (d, *J* = 4.1 Hz, 1H), 6.62 (d, *J* = 4.1 Hz, 1H). $^{13}$C NMR (126 MHz, CDCl$_3$) δ 158.0, 157.6, 155.7, 142.8, 141.1, 139.2, 136.3, 133.0, 131.8, 130.1, 128.9, 128.1, 125.9, 125.1, 124.5, 118.2, 113.0. Calcd. for $C_{19}H_{11}N_3S_2$ (345.44): C, 66.06; H, 3.21; N, 12.16. Found: C, 65.97; H, 3.25; N, 12.22. $\nu$ (DRA, cm$^{-1}$) 2227 (s, C≡N)

**4-(4-(5′-(4-(Diphenylamino)phenyl)-[2,2′-bithiophen]-5-yl)pyrimidin-5-yl)benzonitrile (10)**. The Suzuki cross-coupling reaction of 4-(4-(5′-bromo-[2,2′-bithiophen]-5-yl)pyrimidin-5-yl)benzonitrile (**9**) with 4-(diphenylamino)phenylboronic acid pinacol ester (**5**) (which has been performed according to the general procedure) gave after purification by column chromatography 383 mg (65%) of **10** as a bright yellow solid. Melting point: 172–174 °C. $^1$H NMR (500 MHz, CDCl$_3$) δ 9.16 (s, 1H), 8.51 (s, 1H), 7.84 (d, *J* = 7.9 Hz, 2H), 7.57 (d, *J* = 7.9 Hz, 2H), 7.45 (d, *J* = 8.3 Hz, 2H), 7.31–7.26 (m, 5H), 7.14 (t, *J* = 7.3 Hz, 5H), 7.07 (dd, *J* = 8.2, 5.2 Hz, 4H), 6.96 (d, *J* = 4.1 Hz, 1H), 6.64 (d, *J* = 4.1 Hz, 1H). $^{13}$C NMR (126 MHz, CDCl$_3$) δ 158.0, 157.6, 155.7, 147.8, 147.3, 145.0, 143.0, 141.2, 138.8, 134.5, 133.0, 131.9, 130.1, 129.3, 128.9, 127.3, 126.5, 126.1, 124.7, 124.1, 123.3, 123.2, 123.0, 118.2, 113.0. Calcd. for $C_{37}H_{24}N_4S_2$ (588.74): C, 75.48; H, 4.11; N, 9.52. Found: C, 75.36; H, 3.94; N, 9.56. $\nu$ (DRA, cm$^{-1}$) 2229 (s, C≡N)

4-(4-(3′-Hexyl-[2,2′-bithiophen]-5-yl)pyrimidin-5-yl)benzonitrile (12). Synthesized by the procedure that we described earlier [10].

4-(4-(4′,5′-*Bis*(4-(diphenylamino)phenyl)-3′-hexyl-[2,2′-bithiophen]-5-yl)pyrimidin-5-yl)benzonitrile (14). Synthesized by the procedure that we described earlier [10].

General procedure for the synthesis of bromothiophenyl-substituted pyrimidines (9 and 13).

*N*-Bromosuccinimide (318 mg, 1.5 mmol in the reaction with **8** or 636 mg, 3.0 mmol in the reaction with **12**) was added to a solution of thiophenyl-substituted pyrimidine (**8** or **12**) (1.0 mmol) in DMF (15 mL). The obtained solution was stirred overnight at room temperature. The reaction mixture was diluted with water. The formed precipitate was filtered off, washed with water, dried and purified by silica gel column chromatography with EtOAc/hexane (1:3 to 1:1, *v/v*) as an eluent to afford the desired bromo-substituted products.

**4-(4-(5′-Bromo-[2,2′-bithiophen]-5-yl)pyrimidin-5-yl)benzonitrile (9)**. Obtained from compound **8**, yield 399 mg (94%), yellow solid, mp 197–199 °C. $^1$H NMR (500 MHz, CDCl$_3$) δ 9.16 (s, 1H), 8.53 (s, 1H), 7.85–7.82 (m, 2H), 7.57–7.55 (m, 2H), 7.00 (s, 2H), 6.89 (d, *J* = 4.1 Hz, 1H), 6.62 (d, *J* = 4.1 Hz, 1H). $^{13}$C NMR (126 MHz, CDCl$_3$) δ 158.0, 157.7, 155.5, 141.5, 141.0, 139.7, 137.77, 133.0, 131.7, 131.0, 130.1, 129.0, 125.1, 124.6, 118.1, 113.1, 112.811. Calcd. for $C_{19}H_{10}BrN_3S_2$ (424.34): C, 53.78; H, 2.38; N, 9.90. Found: C, 53.98; H, 2.28; N, 9.83. $\nu$ (DRA, cm$^{-1}$) 2230 (s, C≡N)

4-(4-(4′,5′-Dibromo-3′-hexyl-[2,2′-bithiophen]-5-yl)pyrimidin-5-yl)benzonitrile (13). Synthesized by the procedure that we described earlier [10].

General procedure for the synthesis of benzamide-substituted pyrimidines (D1, D2 and D3).

A solution of KOH (140 mg, 2.5 mmol) in H$_2$O (1 mL) was added to a solution of benzamide-substituted pyrimidine (**6**, **10** or **14**) (0.5 mmol) EtOH (8 mL). The resulting mixture was stirred under reflux for 10 h and then cooled to room temperature. The obtained precipitate was filtered off, washed with ethanol (10 mL) and water (15 mL) and dried in vacuum.

**4-(4-(5-(4-(Diphenylamino)phenyl)thiophen-2-yl)pyrimidin-5-yl)benzamide (D1)**. Obtained from compound **6**, yield 181 mg (69%), yellow solid, mp 241–242 °C. [1]H [1]H NMR (500 MHz, CDCl$_3$) δ 9.19 (s, 1H), 8.60 (s, 1H), 8.27 (t, *J* = 7.6 Hz, 4H), 7.56 (d, *J* = 7.9 Hz, 2H), 7.53 (d, *J* = 7.9 Hz, 2H), 7.28 (d, *J* = 7.8 Hz, 3H), 7.12 (d, *J* = 7.9 Hz, 4H), 7.09–7.01 (m, 5H), 6.99 (d, *J* = 4.1 Hz, 1H), 6.74 (d, *J* = 4.1 Hz, 1H). [13]C NMR (126 MHz, CDCl$_3$) δ 170.3, 157.6, 150.0, 148.4, 147.1, 141.8, 141.2, 138.8, 132.7, 131.13, 131.07, 130.9, 129.9, 129.5, 129.4, 128.2, 126.8, 124.9, 123.6, 123.3, 122.8. HRMS (APCI): *m/z* calcd for C$_{33}$H$_{25}$N$_4$OS: 525.1744 [M+H]$^+$; found: 525.1747. ν (DRA, cm$^{-1}$) 3033 (m, N–H), 1633 (br, C=O), 1488 (s, N–H), 1444 (s, C–N).

**4-(4-(5′-(4-(Diphenylamino)phenyl)-[2,2′-bithiophen]-5-yl)pyrimidin-5-yl)benzamide (D2)**. Obtained from compound **10**, yield 170 mg (56%), orange solid, mp 228–230 °C. [1]H NMR (600 MHz, DMSO-$d_6$) δ 9.14 (s, 1H), 8.65 (s, 1H), 8.14 (s, 1H), 8.06 (d, *J* = 8.1 Hz, 2H), 7.60–7.54 (m, 4H), 7.52 (s, 1H), 7.40 (s, 2H), 7.33 (dd, *J* = 8.5, 7.3 Hz, 4H), 7.16 (d, *J* = 4.0 Hz, 1H), 7.13–7.02 (m, 6H), 6.99–6.92 (m, 2H), 6.55 (d, *J* = 4.1 Hz, 1H). [13]C NMR (151 MHz, DMSO-$d_6$) δ 167.8, 158.6, 157.8, 155.0, 147.7, 147.2, 144.2, 141.5, 140.2, 139.2, 135.0, 134.4, 132.1, 130.3, 130.1, 129.6, 128.8, 127.24, 127.20, 127.0, 125.3, 124.9, 124.5, 124.1, 123.2. HRMS (ESI): *m/z* calcd for C$_{37}$H$_{26}$N$_4$OS$_2$: 606.1543 [M]$^+$; found: 606.1539. ν (DRA, cm$^{-1}$) 1651 (br, C=O), 1489 (s, N–H), 1447 (s, C–N).

**4-(4-(4′,5′-*Bis*(4-(diphenylamino)phenyl)-3′-hexyl-[2,2′-bithiophen]-5-yl)pyrimidin-5-yl)benzamide (D3)**. Obtained from compound **14**, yield 318 mg (68%), dark yellow solid, mp 134–136 °C. [1]H NMR (600 MHz, DMSO-$d_6$) δ 9.18 (s, 1H), 8.72 (s, 1H), 8.12 (s, 1H), 8.06–8.04 (m, 2H), 7.62–7.60 (m, 2H), 7.49–7.46 (m, 2H), 7.30–7.21 (m, 10H), 7.09–7.06 (m, 3H), 7.01–6.96 (m, 12H), 6.89–6.87 (m, 1H), 6.82–6.81 (m, 1H), 6.78–6.75 (m, 2H), 2.22–2.12 (m, 2H), 1.77–1.75 (m, 1H), 1.33–1.08 (m, 5H), 1.03 (q, *J* = 7.1 Hz, 2H), 0.72 (t, *J* = 7.3 Hz, 3H).

[13]C NMR (151 MHz, DMSO-$d_6$) δ 167.7, 158.7, 157.9, 155.0, 147.5, 147.3, 147.1, 144.0, 143.4, 142.7, 141.0, 140.8, 140.3, 135.2, 134.1, 131.4, 130.6, 130.1, 129.9, 129.8, 129.6, 129.2, 128.7, 127.1, 126.8, 126.4, 125.1, 124.7, 124.5, 124.2, 123.7, 123.2, 122.0, 31.4, 30.8, 29.6, 28.8, 22.5, 14.3. HRMS (APCI): *m/z* calcd for C$_{61}$H$_{52}$N$_5$OS$_2$: 934.3603 [M+H]$^+$; found: 934.3608. ν (DRA, cm$^{-1}$) 2927 (m, N–H), 1673 (br, C=O), 1509 (s, N–H), 1465 (s, C–N).

## 3. Results and Discussion

### 3.1. Synthesis

The sequence of Suzuki cross-coupling reactions and bromination step by *N*-bromosuccinimide were utilized as a convenient procedure to obtain the branched push-pull benzonitrile-substituted pyrimidines (**6**, **10**, and **14**) in good yields. The target synthesis has been started from readily available 4-[4-(5-bromothiophen-2-yl)pyrimidin-5-yl]benzonitrile (**3**) (Schemes 1 and 2) [10]. At the last stage, benzonitriles (**6**, **10** and **14**) have been converted to desired amides **D1**–**D3** as a hydrolysis reaction by the action of potassium hydroxide in ethanol solution. The proof of chemical structural was carried out by [1]H and [13]C NMR, HRMS and elemental analysis, and satisfied with their expected structures. We proposed that introduction of a 3-hexylthiophene linker in the dye **D3** would prohibit dye aggregation and prevent undesired charge recombination, giving the decrease of $V_{OC}$ value and the overall performance of DSSC [1–3].

**Scheme 1.** Synthetic route to the starting compound **3**.

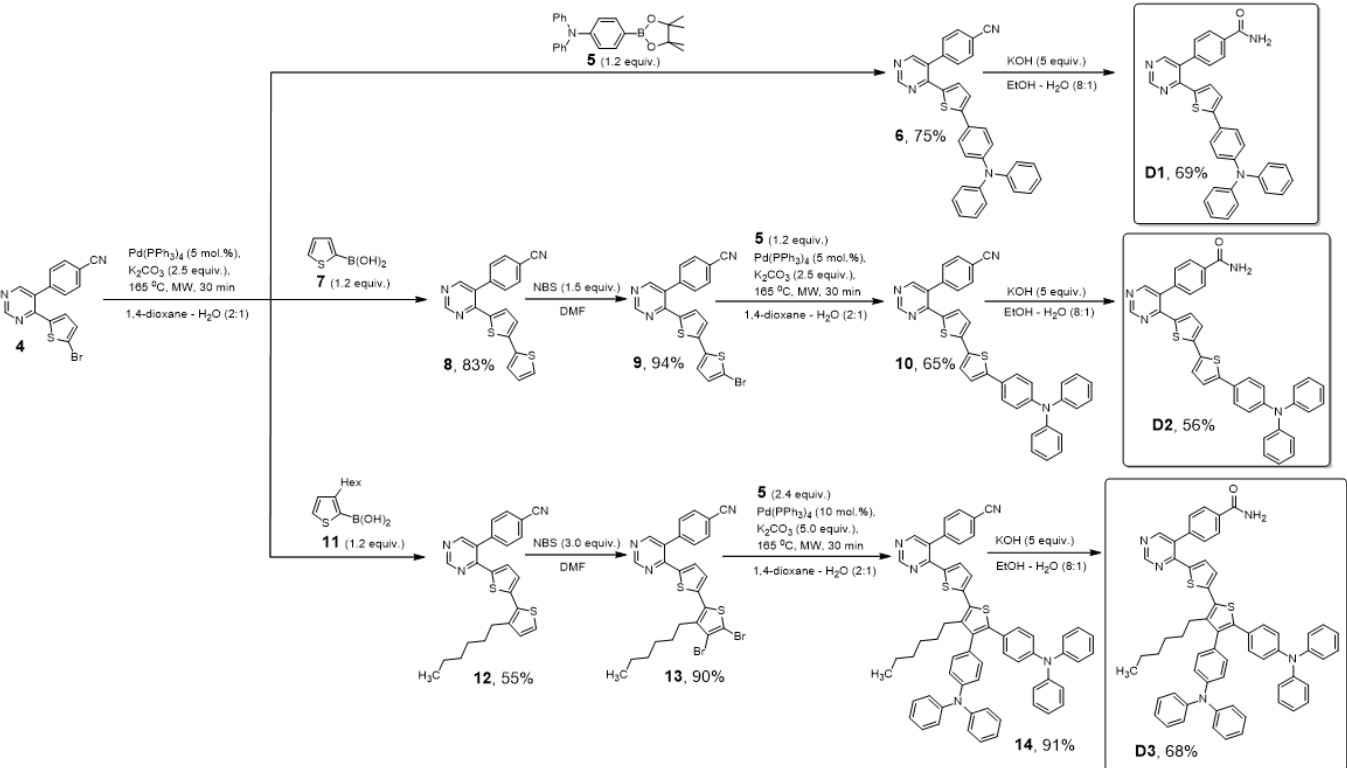

**Scheme 2.** Synthetic route to the dyes **D1–D3**.

### 3.2. Thermal Properties

Thermal stability of synthesized dyes **D1–D3** were investigated using thermo gravimetric analysis (TGA) under air flow. All compounds **D1–D3** were stable up to 388 °C corresponding to a 5% weight loss (Figure 2, Table 1). There was only one step on the TG curve (mass loss for **D1**—18%, for **D2**—30% and for **D3**—48%) for these dyes before 500 °C. As can be seen, the thermal stability of the obtained dyes was appropriate to its applications in DSSCs. The thermal stability in the air decreases in row **D1** > **D2** > **D3**, mostly due to volatility of **D2** and **D3** (some of the dye condensed on cold parts of the TGA furnace, changing their color).

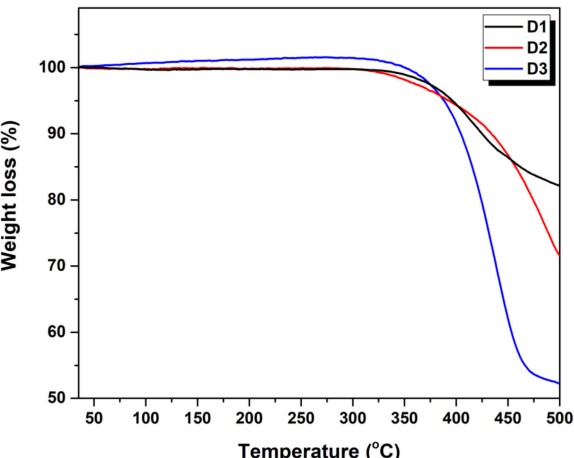

**Figure 2.** TGA curves for the **D1–D3** dyes showing degradation temperature for 5% weight loss in the range of 388–396 °C, which is adequate for device fabrication.

**Table 1.** Photophysical and thermal properties of **D1**–**D3**.

| Dye | Absorption in THF Solution $\lambda$ (nm)/$\varepsilon \cdot 10^{-3}$ ($M^{-1} \cdot cm^{-1}$) | Absorption on $TiO_2$ $\lambda$ (nm) | Photoluminescence | | | $E_{0-0}$ [b] (eV) | $T_d$ [c] (°C) |
|---|---|---|---|---|---|---|---|
| | | | Excitation $\lambda_{max}$ (nm) | Emission $\lambda_{max}$ (nm) | $\Phi_F$ [a] | | |
| **D1** | 407/71.2 300/51.3 | 446 | 407 300 | 520 | 0.62 | 2.67 | 388.3 |
| **D2** | 429/44.9 341/16.8 303/21 | 499 | 429 341 303 | 560 | 0.47 | 2.51 | 393.2 |
| **D3** | 400/19.6 306/59.5 | 433 | 400 306 | 558 | 0.14 | 2.62 | 396.5 |

[a] Fluorescence quantum yield (±10%) determined relative to 3-aminophthalimide in ethanol as standard ($\Phi_F$ = 0.60) [13]. Excitation at 400 nm; [b] the optical band gap $E_{0-0}$ was derived from the intersecting point of absorption and normalized emission spectra in THF; [c] the decomposition temperature corresponding to 5% weight loss from TGA analysis.

Only traces of $[C_4H_4]^+$ and $[C_3H_3N]^+$ ($m/z$ = 52) were observed for **D1** and **D2** in the evolved gases, and no fragments of the dye were observed for D3. The gaseous phase consisted mostly of oxidation products: $CO_2$ ($m/z$ = 44 $[CO_2]^+$, $m/z$ = 12 $[C]^+$ and $H_2O$ ($m/z$ = 18, $[H_2O]^+$, $m/z$ = 17, $[OH]^+$, which appeared prior to $SO_2$ ($m/z$ = 64 $[SO_2]^+$ and $m/z$ = 48 $[SO]^+$).

### 3.3. Photophysical and Electrochemical Properties

The normalized absorption, excitation and emission spectra of the dyes **D1**–**D3** in THF are given in Figure 3, and the corresponding data are summarized in Table 1. The first maximum wavelength absorption in the range 275–340 nm could be assigned the localized $\pi$–$\pi^*$ transitions, while the broad absorption band at 450–550 nm seems to correspond to an intramolecular charge transfer (ICT) from the electron-donating groups to the electron-accepting parts of the push–pull system. It has been found the incorporation of the second 2,5-thienylene-linker into the molecular structure of **D1** led to the bathochromic shift on 22 nm of longwave absorption maximum compared with **D2**, which can be associated with effectively increasing the length of $\pi$-conjugated linker. On contrary, the **D3** containing two triphenylamine groups and additional hexyl substituent have demonstrated a hypsochromic shift of longwave absorption maximum versus **D2** and **D1**. The most plausible explanation for this fact is the interruption of $\pi$-conjugation in the push–pull system due to significant steric hindrances caused by additional alkyl chain and *ortho*-position of two triphenylamine groups.

For the same reasons, fluorescence maxima increased from **D1** to **D3** in a range of 520–560 nm whereas quantum yields dramatically drop from 0.62 to 0.14 (Table 1 and Figure 3b).

The absorption spectra of the dyes adsorbed on the $TiO_2$ nanoparticles are shown in Figure 3a (*on the insert*). The absorption band wavelengths were redshifted by 39 nm for **D1**, 70 nm for **D2** and 33 nm for **D3** relative to those in THF. The great redshift for dyes can be due to the formation of a hydrogen bond between the nitrogen atom of the amide group (or pyrimidine ring) and the hydroxy proton at the $TiO_2$ surface. Moreover, the onsets of the absorption bands are redshifted by 30–70 nm, which is attributable to *J*-type dye's aggregation at the $TiO_2$ electrode [14,15].

To gain further insights into the exact binding modes of the anchoring groups, FTIR spectra were recorded for free dyes **D1**–**D3** and dyes **D1**–**D3** adsorbed $TiO_2$ (Figure 4, Figures S15 and S16). When **D1**–**D3** were adsorbed on the $TiO_2$ nanoparticles, the C=O stretching bands 1633–1673 were slightly shifted around 5–10 $cm^{-1}$, to higher wavenumber compared with the dye powders, that are, the bands can be assigned to the hydrogen-bonded amide group to Brønsted acid sites (Ti–OH species) on the $TiO_2$ surface. Besides the characteristic C=N or C=C bands of the pyrimidine ring were observed at around 1528–1538 and 1564–1588 $cm^{-1}$. When the dyes **D1**–**D3** were adsorbed on the $TiO_2$ surface,

these stretching hands have disappeared and a new strong band appeared at around 1590–1592 cm$^{-1}$, which can be assigned to pyrimidyl groups coordinated to the Lewis acid sites (Ti$^{4+}$) on the TiO$_2$ surface [8]. Thus, the coordination of the dye with the surface of the titanium (IV) oxide could belong to both the interaction with the amide group and with the pyrimidine ring.

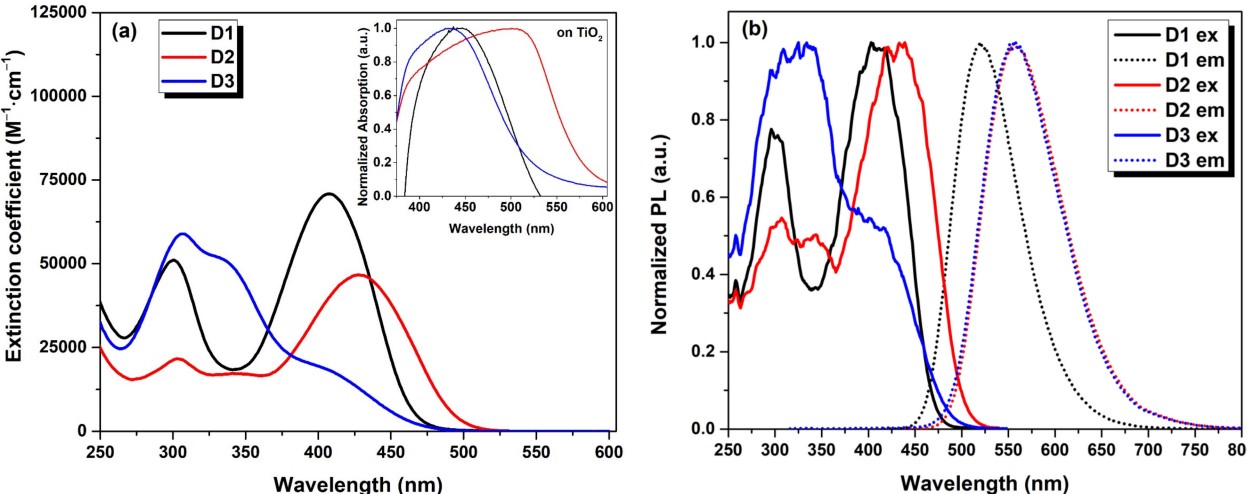

**Figure 3.** (**a**) Absorption spectra of **D1**–**D3** in THF and adsorbed on TiO$_2$ nanoparticles (*in the insert*). (**b**) Emission spectra of compounds **D1**–**D3** in THF.

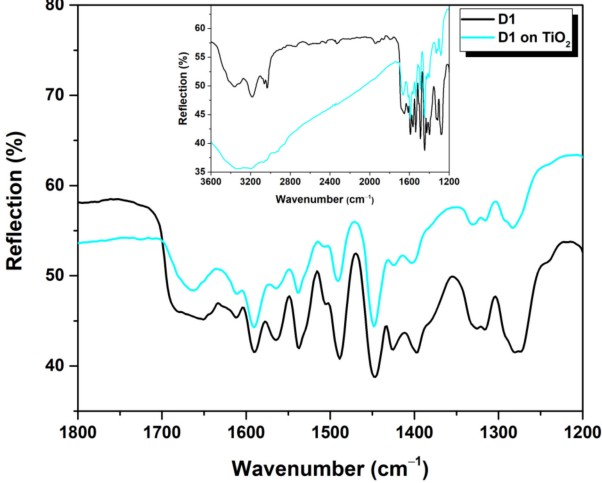

**Figure 4.** FTIR spectra of dye powder and dye adsorbed on TiO$_2$ nanoparticles for **D1**.

To study the electrochemical behavior of synthesized dyes **D1**–**D3** we applied cyclic voltammetry (CV) (Table 2, Figure 5). The cyclic voltammograms shown in Figure 5 demonstrate reversible oxidation and reduction processes for **D1** whereas **D2** and **D3** clearly had a quasi-reversible oxidation process 0.7–1.3 V and a weak reversible reduction process in −1.2 to −1.8 V. On the base of the redox potentials dyes **D1**–**D3**, the HOMO and LUMO energy values were calculated (see Table 2).

**Table 2.** Electrochemical properties of the **D1**–**D3** dyes.

| Compound | $E_{Ox}^{onset}$, V | $E_{Red}^{onset}$, V | $E_{HOMO}$, eV | $E_{LUMO}$, eV | $E_g$, eV |
|:---:|:---:|:---:|:---:|:---:|:---:|
| **D1** | 0.82 | −1.41 | −5.42 | −3.19 | 2.23 |
| **D2** | 0.85 | −1.40 | −5.45 | −3.20 | 2.25 |
| **D3** | 0.89 | −1.23 | −5.49 | −3.37 | 2.12 |

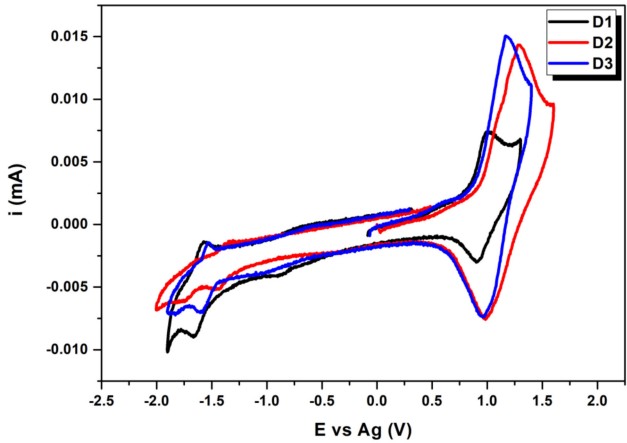

**Figure 5.** Cyclic voltammograms of **D1**–**D3** dyes in CH$_2$Cl$_2$ solutions.

We carried out the comparison of the dye's energy levels with those of the TiO$_2$ conduction band and redox electrolyte to estimate the feasibility of electron injection and regeneration processes (Figure 6). The dye HOMO levels were more positive probably due to that I$^-$ ions (redox level −4.9 eV) can easily reduce the dyes thereby facilitating dye regeneration. On the other hand, the LUMO energies were more negative to the conduction band energy of TiO$_2$ (−4.0 eV) pointing out that excited electrons injection into TiO$_2$ electrode can be spontaneous [8]. It warrants the thermodynamic driving forces for an efficient electron transfer process for electron injection, regeneration and recombination. Thus, these dyes can potentially be used as sensitizers for DSSCs.

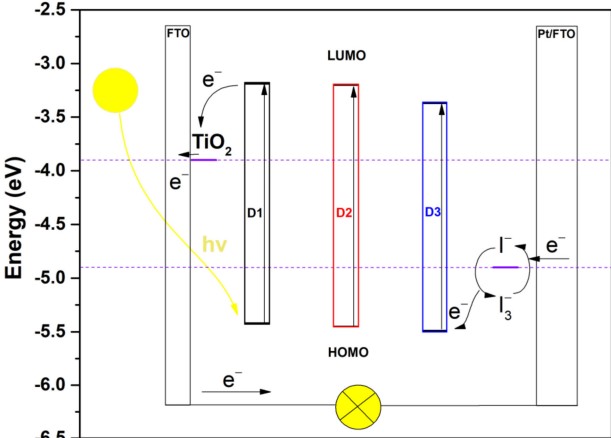

**Figure 6.** Schematic energy diagram for dyes **D1**–**D3**, a nanocrystalline TiO$_2$ electrode and I$_3$$^-$/I$^-$ redox electrolyte.

### 3.4. Photovoltaic Performance of Dyes **D1**–**D3**

The photovoltaic characteristics of DSSCs based on dyes **D1**–**D3** were recorded under simulated solar light (AM 1.5 G, 100 mW/cm$^2$). An image of the sample under study is given as Figure S17. The photocurrent–voltage (*J*–*V*) graph for the cells is displayed in Figure 7 and the numeral results are enumerated in Table 3. As seen from Figure 7 and Table 3, considerable differences were observed in DSSC devices depending on the length of the π-conjugated linker and quantity of triphenylamine electron-donating groups. Undoubtedly, the DSSC based on **D1** demonstrates the highest power conversion efficiency (PCE) of 0.84% among these dyes, with a short-circuit photocurrent density (*J*$_{sc}$) of 2.13 mA·cm$^{-2}$, an open-circuit photovoltage (*V*$_{oc}$) of 0.56 V and a fill factor (*FF*) of 71.85%, respectively. The observed change in a range of *J*$_{SC}$ for DSSCs based on **D1**–**D3** was in good agreement with corresponding changes in the absorption coefficients (see Table 1). Notably,

the DSSCs based on **D2** and **D3** display S-shaped current density–voltage characteristics. It can probably be explained by the existence of charge transport barriers due to the low charge collection efficiency and the fast recombination of electrons in $TiO_2$ with acceptors in the electrolyte [16]. It should be emphasized that the obtained photovoltaic properties lie in a similar range of values typical to other DSSCs based on azines **D4**, **WL101-R** and **OUY-1** (Table 3, Figure 8).

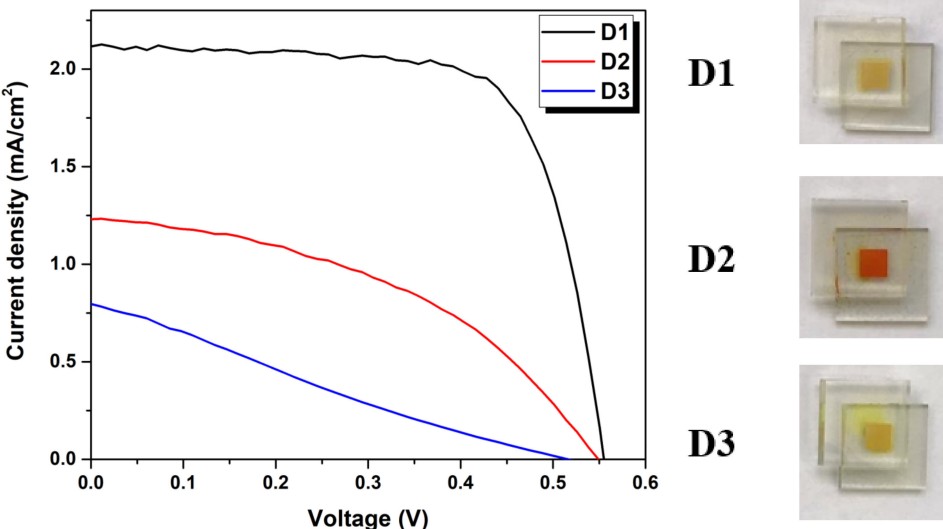

**Figure 7.** The *J–V* curves of and the DSSCs based on dyes **D1**–**D3** measured under simulated AM 1.5G illumination (*Left*). Photos of the devices **D1**–**D3** (*Right*).

**Table 3.** Photovoltaic effectiveness of DSSCs based on dyes **D1**–**D3**.

| Dye | $J_{sc}$ (mA/cm$^2$) | $V_{oc}$ (V) | FF (%) | PCE (%) |
|---|---|---|---|---|
| **D1** [a] | $2.13 \pm 0.12$ | $0.56 \pm 0.01$ | $71.85 \pm 2.72$ | $0.84 \pm 0.02$ |
| **D2** [a] | $1.23 \pm 0.01$ | $0.55 \pm 0.01$ | $43.72 \pm 4.58$ | $0.30 \pm 0.03$ |
| **D3** [a] | $0.78 \pm 0.02$ | $0.51 \pm 0.01$ | $23.14 \pm 0.63$ | $0.09 \pm 0.01$ |
| **WL101-R** [b] | 1.98 | 0.63 | 81.1 | 1.01 |
| **OUY-1** [c] | 3.26 | 0.50 | 58.0 | 0.95 |
| **D4** [d] | 2.04 | 0.52 | 85.0 | 0.91 |

[a] The average performance parameters along with standard deviation errors for 3 parallel devices; [b] literature data [17]; [c] literature data [18]; [d] literature data [9].

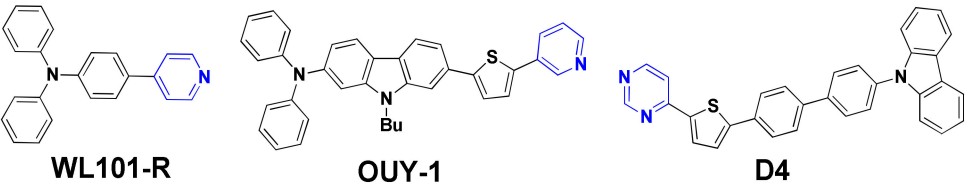

**Figure 8.** Structures of dyes with pyridyl and pyrimidyl anchoring groups [9,17,18].

To gain an understanding of the electron transfer at the interfaces between the sensitizer and the other components such as $TiO_2$, electrolyte and counter electrode, we ran modulation spectroscopy and transient photopotential measurements for photoelectrochemical cells (PECCs) with a photoanode based on the **D1** and **D3** dyes. The transient measurements were performed at a light illumination intensity of 100 mW/cm$^2$ with on–off time intervals of 20 s (Figure 9). The time dependent photopotential graphs (Figure 9a) revealed the changes observed while replacing sensitizer from **D1** to **D3**. The photopotential kept constant for each of the cells for several "on–off" cycles that point out the reproducibility of photopotential generation. It should be emphasized that the sharper photovoltage

decay for **D3** shows that this cell tended to a slightly faster electron recombination rate. Oppositely for the cell based on **D1** the decay rate was a bit slower, which can be ascribed to slower charge recombination on the strength of longer electron lifetime value in this case [19]. Figure 9a also shows that illumination shifted the potential of the photoanode with the **D1** sensitizer by a larger value compared to the **D3** sensitizer. This explains the corresponding increase in the value of $V_{oc}$ in Table 3.

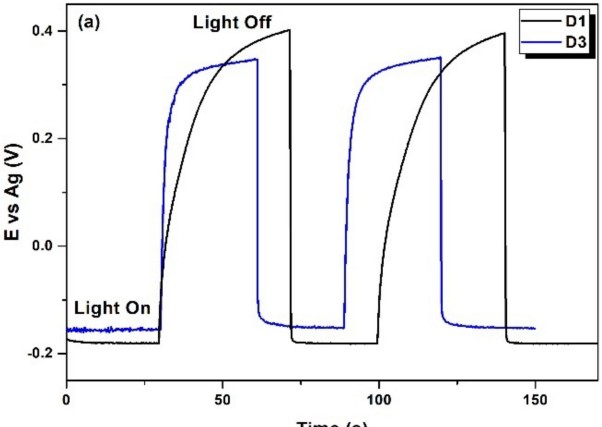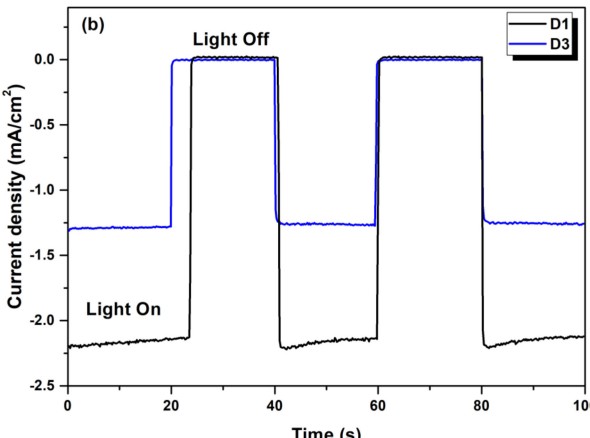

**Figure 9.** Graphs of photopotential (**a**) and photocurrent (**b**) for photoelectrochemical cells (PECC) with photoanodes based on **D1** and **D3** dyes in the dark, and under simulated AM 1.5G illumination (100 mW/cm$^2$).

The transient photocurrent estimation was accomplished under similar conditions (Figure 9b). The results displayed repeating and steady photocurrent response under the given light illumination on–off conditions manifesting repeatability in the assembled PECC. Thus, the effect of the structure of pyrimidine dyes could be evidently noticed in their photovoltaic behavior, especially the modulations caught sight of in the $V_{oc}$ and $J_{sc}$. The photocurrent transient shows a good rectangular shape (Figure 9b). This specifies that there were not diffusion restrictions for **D1**–**D3** dyes regeneration and the imposition of the dark electrochemical processes when using the $I_2/I^-$ mediator system. The electron lifetime ($\tau_{rec}$) and electron transport time ($\tau_{tr}$) are crucial parameters for the description of charge collection efficiency. These can be elucidated by modulation spectroscopy: $\tau_{rec}$ can be measured using intensity modulated voltage spectroscopy (IMVS) under open circuit conditions and $\tau_{tr}$ can be measured using intensity modulated photocurrent spectroscopy (IMPS) under short-circuit conditions [20].

We chose **D1** as the sensitizer with the best PCE and opposite **D3** as the sensitizer with the worst PCE for the investigation of their charge collection efficiency. Here, the IMVS spectra were determined without an external polarization, namely, under the open circuit conditions in the range of 0.5–10 kHz (Figure 10a). Light intensity was controlled with an optical modulator by using a sinusoidal voltage with a frequency, which could be set by an external generator. A little beam modulation depth of approximately 3–5% was selected to avoid non-linear effects. The IMPS spectra were evaluated by using a similar procedure under short-circuit conditions (Figure 10b). The obtained data also indicate the highest charge collection efficiency for the anode fabricated using **D1** with $h_{cc}$ up to 96% (Table 4).

**Table 4.** Data from IMPS and IMVS spectra for the DSSC based on **D1** and **D3** dyes.

| Dye | $f_{tr}$, Hz | $\tau_{tr}$ [a], ms | $f_{rec}$, Hz | $\tau_{rec}$ [b], ms | $h_{cc}$ [c] |
|---|---|---|---|---|---|
| **D1** | 39.5 | 4 | 1.38 | 115 | 0.96 |
| **D3** | 6.34 | 25 | 1.38 | 115 | 0.78 |

[a] $\tau_{tr} = 1/2\pi f_{tr}$, where $f_{tr}$ is the frequency at the minimum of the photopotential spectrum; [b] $\tau_{rec} = 1/2\pi f_{rec}$, where $f_{rec}$ is the frequency at the minimum of the photocurrent spectrum; [c] $h_{cc} = 1 - \tau_{tr}/\tau_{rec}$ is the highest charge collection efficiency.

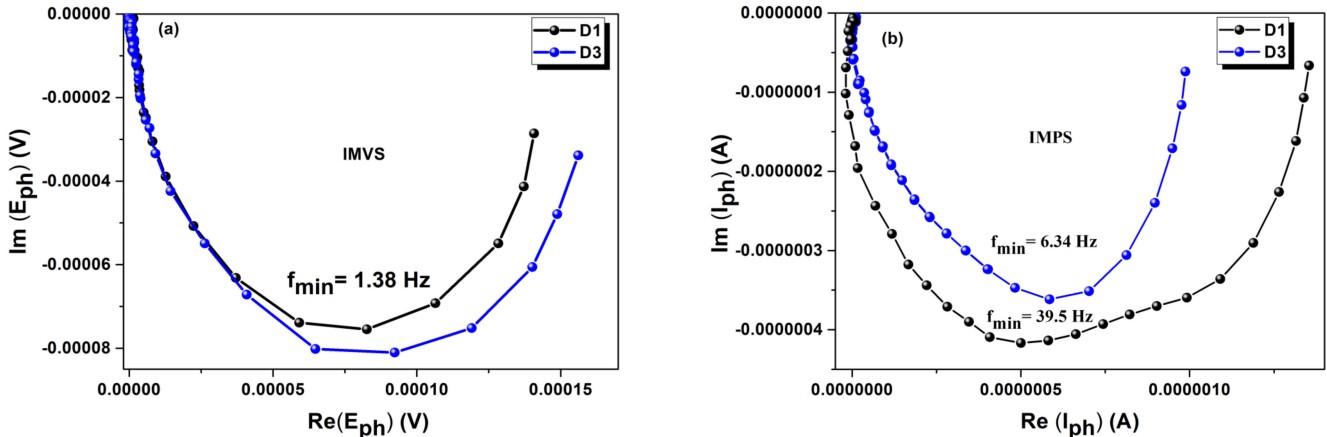

**Figure 10.** IMVS (**a**) and IMPS (**b**) spectra for the DSSC based on **D1** and **D3** dyes.

## 4. Conclusions

Lessons from a new synthetic route for intricately functional dyes based on pyrimidine were taken. In the course, we attempted to find an appropriate functional group namely amide to expand the UV–vis absorption spectra and enhance coordination on $TiO_2$ nanoparticles, to improve the power conversion efficiency of DSSC. Basic photophysical and electrochemical properties of the three new dyes were investigated. Among the fabricated devices, a device **D1** based on the triphenylamine as a donor and an amide group as the acceptor with only one 2,5-thienylene-linker exhibited the highest power conversion efficiency of 0.84% with 2.13 mA/cm² short-circuit current density, 560 mV open-circuit voltage and a fill factor of 71.85%. Although the efficiency was relatively low, we had no doubt that this type of conjugated dyes is a potential sensitizer for semitransparent solar cells that can be easily fabricated by simplifying the active layers and using transparent top electrodes and they would be appealing for applications in building integrated photovoltaics (BIPVs) as electricity-generating facades, shelters, roofs, and windows.

**Supplementary Materials:** The following are available online at https://www.mdpi.com/article/10.3390/electronicmat2020012/s1. Figure S1. ¹H NMR (500 MHz, $CDCl_3$) spectrum of **6**; Figure S2. ¹³C NMR (126 MHz, $CDCl_3$) spectrum of **6**; Figure S3. ¹H NMR (500 MHz, $CDCl_3$) spectrum of **8**; Figure S4. ¹³C NMR (126 MHz, $CDCl_3$) spectrum of **8**; Figure S5. ¹H NMR (500 MHz, $CDCl_3$) spectrum of **9**; Figure S6. ¹³C NMR (126 MHz, $CDCl_3$) spectrum of **9**; Figure S7. ¹H NMR (500 MHz, $CDCl_3$) spectrum of **10**; Figure S8. ¹³C NMR (126 MHz, $CDCl_3$) spectrum of **10**; Figure S9. ¹H NMR (500 MHz, $CDCl_3$) spectrum of **D1**; Figure S10. ¹³C NMR (126 MHz, $CDCl_3$) spectrum of **D1**; Figure S11. ¹H NMR (600 MHz, DMSO-*d6*) spectrum of **D2**; Figure S12. ¹³C NMR (151 MHz, DMSO-*d6*) spectrum of **D2**; Figure S13. ¹H NMR (600 MHz, DMSO-*d6*) spectrum of **D3**; Figure S14. ¹³C NMR (151 MHz, $CDCl_3$) spectrum of **D3**; Figure S15. FTIR spectra of dye powder and dye adsorbed on $TiO_2$ nanoparticles for **D2**; Figure S16. FTIR spectra of dye powder and dye adsorbed on $TiO_2$ nanoparticles for **D3**; Figure S17. Photos of the device **D1** fixed in the measuring stand.

**Author Contributions:** Conceptualization, V.N.C., J.M.N. and G.L.R.; device fabrication, A.S.S. and E.V.K.; methodology, A.R.T., S.A.K., V.A.G., V.V.E., A.S.S. and E.V.V.; validation, E.F.Z., E.V.B. and P.I.L.; formal analysis, A.R.T. and J.M.N.; writing—original draft preparation, E.V.V., A.S.S. and E.V.B.; writing—review and editing, J.M.N. and V.N.C.; supervision, V.N.C.; funding acquisition, G.L.R., E.V.V., V.V.E., S.A.K. and E.V.K. All authors have read and agreed to the published version of the manuscript.

**Funding:** This work (optical and electrochemical properties) was supported by the Russian Foundation for Basic Research (Project No. 18-29-23045 mk). E.V.V. is grateful to the financial support for the synthetic part from the Ministry of Education and Science of the Russian Federation within the framework of the State Assignment for Research (Project No. AAAA-A19-119011790132-7). NMR experiments were carried out by using equipment of the Center for Joint Use «Spectroscopy and Analysis of Organic Compounds» at the Postovsky Institute of Organic Synthesis of the Ural Branch

**Institutional Review Board Statement:** Not applicable.

**Informed Consent Statement:** Not applicable.

**Data Availability Statement:** The data presented in this study are available on request from the corresponding authors.

**Conflicts of Interest:** The authors declare no conflict of interest.

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
