# Peer review of "Pyrimidine-Based Push–Pull Systems with a New Anchoring Amide Group for Dye-Sensitized Solar Cells"

_electronicmat, doi:10.3390/electronicmat2020012_

Round 1
Reviewer 1 Report
The manuscript may be published after revision. My major criticisms are listed below.
- The emission spectra are reported in figure 3b and the data in table 1 but are not discussed in the text.
- The authors complain that “there have been only several examples of azine-functionalized push-pull systems for DSSC applications, and efficiencies with such dyes have been low. Particularly, in our recent work for the first time, we synthesized a series of novel push-pull dyes bearing pyrimidine as the anchoring group. Unfortunately, it has been found that DSSCs based on these dyes had unimpressive power conversion efficiency (PCE) up to 1 %.”. However, D1 displayed PCE < 1.
- The author should explain why they introduced an additional 3-hexyl substituent in D3 instead of, for instance, the second triphenylmine in 3-position, probably avoiding the interruption of π-conjugation in the push-pull system due to steric hindrance.
Author Response
1. The emission spectra are reported in figure 3b and the data in table 1 but are not discussed in the text.
The answer:
The discussion of the emission spectra for dyes D1-D3 has been added.
"For the same reasons, fluorescence maxima increased from D1 to D3 in a range of 520-560 nm whereas quantum yields dramatically drop from 0.62 to 0.14 (Table 1 and Fig. 3b)."
2. The authors complain that “there have been only several examples of azine-functionalized push-pull systems for DSSC applications, and efficiencies with such dyes have been low. Particularly, in our recent work for the first time, we synthesized a series of novel push-pull dyes bearing pyrimidine as the anchoring group. Unfortunately, it has been found that DSSCs based on these dyes had unimpressive power conversion efficiency (PCE) up to 1 %.”. However, D1 displayed PCE < 1.
The answer:
The reviewer is right. When we started this research, we didn't know what will be the PCEs for new DSSCs. Unfortunately, the introduction of the auxiliary amide anchoring group does not increase PCE in our case but we believe that it is only the first example of photosensitizers with this type of anchoring and we expect more success in the near future.
3. The author should explain why they introduced an additional 3-hexyl substituent in D3 instead of, for instance, the second triphenylmine in 3-position, probably avoiding the interruption of π-conjugation in the push-pull system due to steric hindrance.
The answer:
We proposed that introduction of a 3-hexylthiophene linker in the dye D3 would prohibit dye aggregation and prevent undesired charge recombination, which causes the decrease of VOC value and the overall performance of DSSCs [1-3].

Reviewer 2 Report
- The authors have to add more citations. In its current form, it's too less.
- In this study, how many devices were checked? The reviewers suggest that any of the tables, can add the standard deviation, and the electrical properties should also be compared with other publications for detailed investigation.
-
Please provide the photo of the fabricated devices for electrical measurement
Author Response
1. The authors have to add more citations. In its current form, it's too less.
The answer:
The authors believe that the references provided are sufficient to characterize the research. However, a few extra references have been added.
2. In this study, how many devices were checked? The reviewers suggest that any of the tables, can add the standard deviation, and the electrical properties should also be compared with other publications for detailed investigation.
The answer:
The average performance parameters along with standard deviation errors are given for 3 parallel devices. The standard deviation for fluorescence quantum yields was added whereas the standard deviation for photovoltaic systems was given in the original version of the paper (see Table 3).
3. Please provide the photo of the fabricated devices for electrical measurement
The answer:
The photo of the fabricated devices for electrical measurements was displayed in the original version of the paper in Figure 7 (right) whereas another photo of the device D1 attached to the measuring stand was added in Figure S17.

Reviewer 3 Report
This paper is the study of new donor-π-acceptor pyrimidine-based dyes comprising an amide moiety as an anchoring group. It is interesting and could be accepted for publication after minor revised. The comments are as follows:
- The photovoltaic performance of DSSCs based on D1-D3 are very low. The authors should explain why these dyes can be applied on DSSCs in the future.
- In Fig. 8, the data of D2 are missing. The authors should provide D2 results and compare it with D1 and D3.
- The meanings of Figure 9 should be discussed furthermore in this manuscript.
Author Response
- The photovoltaic performance of DSSCs based on D1-D3 are very low. The authors should explain why these dyes can be applied on DSSCs in the future.
The answer:
The authors agree with the Reviewer. Of course, the dyes D1-D3 cannot be utilized directly but we believe that any further modifications of dyes with an amide anchoring group allow in the future to fabricate DSSCs with much higher performance.
- In Fig. 8, the data of D2 are missing. The authors should provide D2 results and compare it with D1 and D3.
The answer:
Unfortunately, we cannot present the results for the D2 dye neither in Figure 8 nor in Figure 9. These data were not initially obtained, because we took as a basis only the extreme solar cells with the best D1 and the worst D3 efficiency indicators. The corresponding rationale was in the original text of the article (violet highlighted):
" To gain an understanding of the electron transfer at the interfaces between the sensitizer and the other components such as TiO2, electrolyte, and counter electrode, we ran modulation spectroscopy and transient photopotential measurements for photoelectrochemical cells (PECC) with a photoanode based on the D1 and D3 dyes. ..."
- The meanings of Figure 9 should be discussed furthermore in this manuscript.
The answer:
Figure 9 was discussed in the original version of the paper (violet highlighted):
" We have chosen D1 as sensitizer with the best PCE and opposite D3 as sensitizer with the worst PCE for the investigation of their charge collection efficiency. Here, the IMVS spectra have been determined without an external polarization, namely, under the open circuit conditions in the range of 0.5–10 kHz (Fig. 9a). Light intensity was controlled with an optical modulator by using a sinusoidal voltage with a frequency, which could be set by an external generator. A little beam modulation depth of approximately 3–5% was selected to avoid non-linear effects. The IMPS spectra were evaluated by using a similar procedure under short-circuit conditions (Fig. 9b). The obtained data also indicate the highest charge collection efficiency for the anode fabricated using D1 with hcc up to 96% (Table 4)....."

Reviewer 4 Report
Below are some issues that need to be considered:
1.Missing EQE spectra. The EQE results should be integrated into a resulting Jsc that can be compared with IV measured data.
2. All figures must be numbered consecutively. Page 10 in the Supplementary Material, figure number should be 11 instead of 31. The remaining figures should be renumbered accordingly. Figure numbers should be labeled according to their sequence in the text.
Author Response
1.Missing EQE spectra. The EQE results should be integrated into a resulting Jsc that can be compared with I-V measured data.
The answer:
Unfortunately, EQE spectra for solar cells were not recorded because we did not have the appropriate measuring equipment and we could not travel to another institute owing to COVID restrictions. Currently, a device for measuring EQE (SCS10-PEC Photoelectrochemistry Measurement System) has been ordered but it will arrive in the laboratory by the end of this summer. Nevertheless, we believe that the absence of the EQE measurement data is not critical, since the Jsc obtained from the EQE integration may confirm that our dyes did not break new efficiency records.
2. All figures must be numbered consecutively. Page 10 in the Supplementary Material, figure number should be 11 instead of 31. The remaining figures should be renumbered accordingly. Figure numbers should be labelled according to their sequence in the text.
The answer:
All figures were renumbered in the Supplementary Material.

Round 2
Reviewer 1 Report
Authors' answers are satisfactory
Reviewer 4 Report
I think the paper now is in an acceptable stage for publication.
Author Response
Many thanks for your good opinion about our research!